# Fear of Falling in Older Adults Treated at a Geriatric Day Hospital: Results from a Cross-Sectional Study

**DOI:** 10.3390/ijerph19148504

**Published:** 2022-07-12

**Authors:** Eva M. Melendo-Azuela, Julia González-Vaca, Eva Cirera

**Affiliations:** 1The Doctoral Programme in Comprehensive Care and Health Services, University of Vic-Central University of Catalonia, 08500 Vic, Spain; 2Faculty of Medicine and Health Sciences, School of Nursing, University of Barcelona, 08007 Barcelona, Spain; 3Nursing Research Group (GRIN) from the IDIBELL Translational Medicine Area, University of Barcelona, 08007 Barcelona, Spain; juliagonzalezvaca@ub.edu; 4Sport and Physical Activity Research Group, Center for Health and Social Research, Department of Experimental Sciences and Methodology, Faculty of Health Sciences and Welfare, University of Vic-Central University of Catalonia, 08500 Vic, Spain; eva.cirera@uvic.cat

**Keywords:** fear of falling, prevalence, determinants, geriatric day hospital, urban area, Activities-Specific Balance Confidence (ABC)

## Abstract

(1) Background: The fear of falling (FOF) is a geriatric syndrome that causes a decrease in daily activities and personal autonomy. Its prevalence is highly variable as are the methodologies used to assess it. This study aimed at estimating the prevalence and describing the main determinants of FOF in older adults attending a geriatric day hospital. (2) Methods: Descriptive, cross-sectional study of individuals aged ≥70 years, who attended an ambulatory functional rehabilitation group in the metropolitan area of Barcelona. FOF was assessed using the Activities-Specific Balance Confidence (ABC) scale. Other recorded outcomes were: sex, age, marital status, living alone, level of education, degree of autonomy, pain, previous falls, visual acuity, and signs of depression. Prevalence was estimated overall and according to the possible determinants. (3) Results: The study included 62 individuals (66.1% women), with a prevalence of fear of falling of 38.7% (95% CI 26.2–51.2%). The identified determinants were pain (OR = 7.4, 95% CI 1.4–39.7), a history of falls (OR = 25.3, 95% CI 2.1–303.4), poor visual acuity (OR = 5.6, 95% CI 1.0–29.8), and signs of depression (OR = 19.3, 95% CI 1.4–264.3). (4) Conclusions: The prevalence and determinants of fear of falling in older adults attending geriatric day hospitals were similar to those described in those dwelling in the community.

## 1. Introduction

The fear of falling syndrome is defined as a decrease in usual daily activities caused by said fear both in people with and without a history of falls [1]. The main consequences of this syndrome are: a decrease in functional autonomy and the ability to perform basic daily-life activities and usual physical activity; an increased risk of falling, depression and progressive loss of the quality of life of older adults [2]. These consequences make individuals frail and vulnerable [3].

The fear of falling syndrome is a significant health problem in older adults living in the community. In this population, the prevalence of fear of falling is highly variable, ranging from 20.8 to 85.0%, based on the study methodology and measurement systems used [2]. In Spain, a prevalence between 31.2 and 71.6% has been described [4,5,6]. On the other hand, the main determinants of fear of falling in older adults living in the community include the female sex, alteration of physical function, use of technical help to walk, a history of falls, frailty, a perception of low quality of health, depression, chronic diseases, limitations of instrumental activities of daily life, pain, level of education, and visual acuity [7,8,9,10].

Some of the older adults living in the community go to a day hospital. For example, in Catalonia, there are around 1.1 million people aged 70 years or more [11], of which around 8000 are treated at the 73 geriatric day hospitals that exist in Catalonia [12,13]. These centers provide ambulatory day care to chronic patients with exacerbations and/or difficulty in managing their pathology through a portfolio of services, including comprehensive and interdisciplinary geriatric care, but also the administration of specific treatments such as cognitive stimulation, functional rehabilitation, or prevention of falls. Patients attending geriatric day hospitals can be derived from an intermediate care center to reduce the time spent in hospital, or from primary care to solve a health need that requires specialized geriatric care.

Several recent studies have described the prevalence of fear of falling and its possible determinants in the community population. However, no specific data for day-hospital patients are available—individuals who, although still living in the community and not requiring hospitalization, need specialized health care. Knowing the prevalence and determinants of fear of falling in this population would allow us to adopt specific guidelines and strategies to prevent fear of falling and improve their care. Therefore, the aim of this study was to estimate prevalence and describe the main determinants of fear of falling in a sample of older adults attending an ambulatory geriatric care center in the urban area of Barcelona.

## 2. Materials and Methods

### 2.1. Study Design and Population

This was a descriptive, cross-sectional study in community-dwelling individuals aged ≥70 years, who attended a functional rehabilitation group at the geriatric day hospital of L’Hospitalet de Llobregat Healthcare Center (Barcelona) between September 2017 and December 2018. Study participants had to be able to walk 10 m without stopping (with or without technical help) and not present with cognitive deterioration (<2 errors in the Pfeiffer scale [14]). Illiterate individuals were excluded due to their inability to complete self-administered questionnaires.

All participants signed an informed consent document. Data were collected by assessing participants’ medical histories and the Activities-Specific Balance Confidence (ABC) questionnaire. If an individual could not complete this questionnaire on their own, he/she could get help from relatives or the nurse on duty. The study protocol was approved by the Clinical Research Ethics Committee of the University Hospital of Bellvitge (reference PR269/17].

### 2.2. Study Outcomes

The primary outcome of the study was fear of falling, measured using the Activities-Specific Balance Confidence (ABC) Scale (reliability 0.96, validity 0.84, sensitivity 0.65), a 16-item questionnaire scored from 0 to 100 measuring degree of confidence of balance in not falling while performing activities of daily living [15]. An individual is considered to have a fear of falling when his/her score is lower than 67 points [16].

Other variables analyzed were gender, age, marital status, living alone and level of education. Degree of autonomy was analyzed based on the Barthel Index (<21 points, total dependency; 21–60, severe dependency; 61–90, moderate dependency; 91–99, slight dependency; and 100, independence) [17]. Similarly, participants’ pain was assessed using the numerical rating scale (NRS): 0 points, no pain; 1–3, mild pain; 4–7, moderate pain; ≥8, severe pain [18]. To dichotomize this outcome, the values 0–3 and 4–10 were used to define the absence and presence of pain, respectively. The history of falls in the last six months was assessed by asking the direct question, “Have you fallen in the last six months?” or by checking participants’ medical histories; visual acuity was assessed using the Jaeger chart (<7 points, correct; ≥7 points, incorrect) [19] and depression was assessed using a short version of the Yesavage scale (0–1 points, no signs of depression; ≥2 points, presence of signs of depression) [20]. All these variables were collected from participants’ medical histories or by asking them.

### 2.3. Statistical Analysis

Quantitative variables were described using medians and interquartile ranges (IQRs), and categorical variables were described by absolute and relative frequencies. The prevalence of fear of falling was estimated using a 95% confidence interval (95% CI).

To identify the determinants of fear of falling, a bivariate analysis with logistic regression was performed with the following variables: sex, age, marital status, pain, previous falls in the last six months, signs of depression, degree of autonomy, and visual acuity. Then, a multivariate logistic regression model was carried out, with fear of falling as the dependent variable and, as independent variables, those with a *p*-value < 0.25 of Wald test in the bivariate analysis, or those that, when excluded from the model, the estimated coefficients for the remaining variables changed markedly in magnitude (>10%) [21]. Given the low number of cases, it was decided not to assess the interactions between possible determinants. Results were presented as the estimated prevalence of fear of falling and the corresponding OR for each category, together with their respective 95% CI. The level of statistical significance was set at a bilateral alpha value of 0.05. All statistical analyses were performed using the SPSS statistics software for Windows, version 22.

## 3. Results

### 3.1. Study Participants’ Characteristic

The study included a total of 62 individuals, whose socio-demographic and clinical characteristics are shown in Table 1. Approximately two out of three participants were women, and more than half were aged between 75 and 84 years. Most participants had a primary education, lived with someone, and were either married or widowed. Over 60% were independent and did not present with pain, and about one in four had poor visual acuity.

### 3.2. Prevalence of Fear of Falling

The median (IQR) of the ABC scale was 76.57 (37.97). The prevalence of fear of falling in the population sample under study was 38.7% (95% CI 26.2–51.2%). Based on sex, the prevalence in women was 36.6% (95% CI 21.0−52.0%) and, in men, 42.9% (95% CI 20.0−66.0%).

Table 2 shows participants’ socio-demographic and clinical characteristics based on whether they had a fear of falling or not. Regarding individuals with fear of falling, 25% were over 84 years old, 50% were independent and 54.2% reported no pain. In addition, 25% had fallen in the last six months, and 33.3% had poor visual acuity and showed signs of depression 37.5%. Regarding individuals without fear of falling, about one in six was over 84 years old, more than 60% were independent, and over 75% reported no pain. Approximately 8% had fallen in the last six months, and 21.1% had poor visual acuity and 3% showed signs of depression.

### 3.3. Determinants of Fear of Falling

Table 3 and Table 4 show the influence of the different outcomes on the percentage of participants with fear of falling. Results from the bivariate analysis showed that the outcomes associated with a higher prevalence of fear of falling were: presence of pain, falls in the last six months, and presence of signs of depression (Table 3). These outcomes remained significant in the multivariate model (Table 4), in a way that individuals that were more likely to have a fear of falling were those that reported pain (OR = 7.42), had fallen in the last six months (OR = 25.33), showed signs of depression (OR = 19.33), or had problems of visual acuity (OR = 5.56).

## 4. Discussion

In this study, we observed that the prevalence of fear of falling in individuals aged 70 years or more attending geriatric day hospitals was 38.7%. Those outcomes associated with a higher probability of the fear of falling syndrome were: previous falls, presence of signs of depression, presence of pain, and poor visual acuity.

Although studies of the prevalence of fear of falling in older adults yield highly heterogeneous results, our study showed a similar prevalence to that described by other authors in community-dwelling populations [4,5,8,22]. From the start, patients attending day hospitals might be expected to have a higher prevalence of fear of falling than community-dwelling older adults, and a similar prevalence to that observed in pre-frail and frail populations [23,24]; instead, we found a similar prevalence to that of the general population. However, although prevalence scores are similar, we must understand the use of day hospitals as an advantage for these individuals. Having them in the facility for some time gives us the possibility to detect the fear of falling and perform interventions on the identified determinants to reduce this fear.

The variability in the prevalence of fear of falling might be explained through the measurement method. Most prevalence studies are based on a direct question with different levels of responses, consisting of 3–5 points on a Likert scale [4,5,8,22,25,26]; however, Thiamwong and Suwanno considered the use of one simple question as a limitation, since it does not discuss the multifactorial nature of the fear of falling [22]. Other authors use a validated measurement scale, namely the Falls Efficacy Scale (FES) [6,27], which only assesses indoor activities and, therefore, usually applies to individuals with limitations or low mobility. There is a modified version of FES (mFES) expanding four items referring to activities in the open air. Another scale used is the ABC that measures balance confidence in not falling while performing these activities; it is extended to add a further six items related to the instrumental activities of daily living included in the FES, so it can be applied to individuals with more functionality. Due to the type of patients attending day hospitals, we considered it more appropriate to use the ABC scale in our study. According to the systematic review of Alarcón et al., the prevalence of fear of falling is higher in studies using the FES scale than in those using the ABC scale [28].

Regarding the determinants of fear of falling, our results showed a higher probability of fear of falling in those individuals who had fallen in the last six months and in those who presented with signs of depression, which is consistent with results from previous studies performed in the general population [8,25,26,27]. Likewise, the probability of fear of falling was higher in individuals with sight problems and in those that reported pain, as described by Liu [27] and Stubbs et al. [9], respectively. However, unlike other authors [4,5,6,8,22,26,27], we did not observe any association between fear of falling and socio-demographic characteristics such as age, level of education, marital status, or gender. In scientific literature, sex is one of the most common determinants of fear of falling, and several studies have shown a higher prevalence of fear of falling in women than in men, being twice as much, in some cases [4,5,6,8,22,26,27]. Some authors have attributed this difference to women’s higher concern with their health [29] or a higher tendency to develop osteoporosis or a weaker musculoskeletal system [30]. However, in our population, the prevalence of fear of falling was similar in both sexes since, compared with other studies, the prevalence in women was lower and that in men was higher [4,5,6,8,22,26,27]. In this sense, using patients treated in a specialized geriatric center as a starting point, it is possible that men in our study were as concerned about their health and the consequences of a fall as women.

To our knowledge, it is the first time that the prevalence and determinants of fear of falling are described in older adults, who, although still part of the community, already require assessment and follow-up in a specialized healthcare setting. Initially, we expected that a person who uses a geriatric health service may have a pre-fragility situation and probably have higher prevalence of fear of falling, but we observed that they have a similar situation to older people who live in the community, so it can be concluded that the profile of the person treated in a geriatric day hospital is similar to other older people who do not attend such a resource. In addition, when using a validated scale to measure fear of falling, not only do we assess its presence, but also the limitations entailed in the performance of normal activities, since scales measure an individual’s confidence to avoid falling during the performance of daily-life activities. However, our study has some limitations. Being a cross-sectional study, we were not able to identify possible determinants of fear of falling; we were only able to analyze the influence of those determinants that were already described in the literature. In addition, the sample of participants was too small to estimate the significance of the effect of the determinants with enough accuracy, obtaining very wide confidence intervals. Another limitation not measured was chronic potential correlates of fear of falling.

Adults attending geriatric day hospitals are followed up between two and three months, which represents an opportunity to screen fear of falling and perform the necessary intervention. Therefore, it would be convenient to include a validated scale of fear of falling within the comprehensive geriatric assessment that is performed on patients of day hospitals, to detect it as early as possible and try to improve their efficacy in their daily lives. Likewise, programs intended to prevent and treat the determinants of fear of falling in geriatric day hospitals with the final objective of preventing dependency and a decrease in quality of life should be applied.

Fear of falling can be prevented with physical activity programs that improve walking and increase the level of confidence to avoid falls during daily-life activities [31]. In fact, a proper physical activity program may be efficient, even in frail, older adults [32]. In addition, due to the multifactorial nature of fear of falling, the effectiveness of these programs improves when combined with health education programs [33], so they should be applied together to improve individuals’ confidence in the performance of daily activities and prevent falls. With respect to pain, it might be necessary to perform interventions to reduce it through pharmacological and non-pharmacological treatments [34]. It would also be advisable to use screening methods for depression and sight defects in order to refer the individual to the necessary professional or service. On the other hand, although it is not an outcome analyzed in this study, polypharmacy should be assessed due to its potential effects in older adults [35].

## 5. Conclusions

In conclusion, in our study, more than a third of older adults living in the community and attending a geriatric day hospital had a fear of falling, which may negatively impact their quality of life. The prevalence and determinants of fear of falling in patients attending day hospitals were similar to those in community-dwelling older adults. Future studies should analyze the possible association of fear of falling with frailty and, particularly, if fear of falling may be a predictor of frailty.

## Figures and Tables

**Table 1 ijerph-19-08504-t001:** Socio-demographic and clinical characteristics of study participants, *n* (%)*. n* = 62.

Gender	
Female Sex	41 (66.1)
Male Sex	21 (33.9)
**Age**	
70–74 years	15 (24.2)
75–84 years	35 (56.5)
>84 years	12 (19.4)
**Level of education**	
Primary (not completed)	4 (6.5)
Primary	57 (91.9)
Secondary	1 (1.6)
**Marital status**	
Widowed	28 (45.2)
Married	32 (51.6)
Divorced	2 (3.2)
Lives alone	23 (37.1)
**Degree of autonomy ^a^**	
Independence	38 (61.3)
Slight dependency	10 (16.1)
Moderate dependency	14 (22.6)
Severe dependency	0 (0.0)
Total dependency	0 (0.0)
**Pain ^b^**	
No pain	43 (69.4)
Mild pain	2 (3.2)
Moderate pain	12 (19.4)
Severe pain	5 (8.1)
**Falls in the last six months**	9 (14.5)
**Poor visual acuity ^c^**	16 (25.8)
**Signs of depression ^d^**	10 (16.1)

^a^ Based on the Barthel Index score (100, independence; 91–99, slight dependency; 61–90, moderate dependency; 21–60, severe dependency; <21, total dependency). ^b^ Based on the numerical rating scale (NRS) (0, no pain; 1–3, mild pain; 4–7, moderate pain; ≥8, severe pain). ^c^ Score ≥ 7 on the Jaeger chart (≥7). ^d^ Score ≥ 2 on the short version of the Yesavage scale.

**Table 2 ijerph-19-08504-t002:** Socio-demographic and clinical characteristics of study participants based on the presence of fear of falling, *n* (%).

	Fear of Falling (*n* = 24)	No Fear of Falling (*n* = 38)
**Gender**		
Female sex	15 (62.5)	26 (68.4)
Male sex	9 (37.5)	12 (31.6)
**Age**		
70–74 years	6 (25.0)	9 (23.7)
75–84 years	12 (50.0)	23 (60.5)
>84 years	6 (25.0)	6 (15.8)
**Level of education**		
Primary (not completed)	1 (4.2)	3 (7.9)
Primary	22 (91.7)	35 (92.1)
Secondary	1 (4.2)	0 (0.0)
**Marital status**		
Widowed	12 (50.0)	16 (42.1)
Married	10 (41.7)	22 (57.9)
Divorced	2 (8.3)	0 (0.0)
Lives alone	9 (37.5)	14 (36.8)
**Degree of autonomy ^a^**		
Independence	12 (50.0)	26 (68.4)
Slight dependency	5 (20.8)	5 (13.2)
Moderate dependency	7 (29.2)	7 (18.4)
Severe dependency	0 (0.0)	0 (0.0)
Total dependency	0 (0.0)	0 (0.0)
**Pain ^b^**		
No pain	13 (54.2)	30 (78.9)
Mild pain	0 (0.0)	2 (5.3)
Moderate pain	7 (29.2)	5 (13.2)
Severe pain	4 (16.7)	1 (2.6)
**Falls in the last six months**	6 (25.0)	3 (7.9)
**Poor visual acuity ^c^**	8 (33.3)	8 (21.1)
**Signs of depression ^d^**	9 (37.5)	1 (2.6)

^a^ Based on the Barthel Index score (100, independence; 91–99, slight dependency; 61–90, moderate dependency; 21–60, severe dependency; <21, total dependency). ^b^ Based on the numerical rating scale (NRS) (0, no pain; 1–3, mild pain; 4–7, moderate pain; ≥8, severe pain). ^c^ Score ≥ 7 on the Jaeger chart (≥7). ^d^ Score ≥ 2 on the short version of the Yesavage scale.

**Table 3 ijerph-19-08504-t003:** The influence of possible determinants on the prevalence of fear of falling.

	Prevalence (%)	95% CI	OR	95% CI	*p*
**Gender**					
Male	42.86	(19.77–65.94)	1.00		
Female	36.59	(21.19–51.98)	0.83	(0.28–2.45)	0.740
**Age**					
<75	40.00	(11.92–68.08)	1.00		
75–84	34.29	(17.74–50.83)	0.70	(0.20–2.47)	0.575
>84	50.00	(16.82–83.13)	1.6	(0.33–7.85)	0.562
**Marital status ^a^**					
Married	31.25	(14.27–48.23)	1.00		
Widowed	42.86	(23.32–62.40)	1.89	(0.64–5.52)	0.247
**Lives alone**					
No	38.46	(22.48–54.44)	1.00		
Yes	39.13	(17.55–60.71)	1.22	(0.41–3.64)	0.715
**Degree of autonomy ^b^**					
Independence	31.58	(16.1–47.06)	1.00		
Slight dependency	50.00	(12.3–87.70)	2.60	(0.59–11.49)	0.206
Moderate dependency	50.00	(20.04–79.96)	2.08	(0.59–7.30)	0.251
**Pain ^c^**					
No	28.89	(15.12–42.66)	1.00		
Yes	64.71	(39.38–90.03)	4.60	(1.38–15.20)	**0.013** *****
**Falls in the last six months**					
No	38.71	(20.55–56.87)	1.00		
Yes	50.00	(16.82–83.13)	5.67	(1.04–31.00)	**0.045** *****
**Correct visual acuity ^d^**					
Yes	34.78	(20.48–49.08)	1.00		
No	50.00	(22.48–77.52)	2.50	(0.74–8.50)	0.141
**Signs of depression ^e^**					
No	28.85	(16.11–41.58)	1.00		
Yes	90.00	(67.38–112.62)	21.00	(2.44–180.77)	**0.006** *****

^a^ Divorced individuals were excluded from the analysis due to the low number of cases. ^b^ Based on the Barthel Index score (100, independence; 91–99, slight dependency; 61–90, moderate dependency; 21–60, severe dependency; <21, total dependency). ^c^ Based on the numerical rating scale (NRS) (0–3, absence of pain; 4–10, presence of pain). ^d^ Score ≥ 7 on the Jaeger chart (≥7). ^e^ Score ≥ 2 on the short version of the Yesavage scale. * *p* < 0.05.

**Table 4 ijerph-19-08504-t004:** Adjusted model of the prevalence of fear of falling.

	OR	95% CI	*p*
**Marital status ^a^**			
Married	1		
Widowed	4.62	(0.88–24.27)	0.071
**Pain ^b^**			
No	1		
Yes	7.42	(1.39–39.69)	**0.019** *****
**Falls in the last six months**			
No	1		
Yes	25.33	(2.12–303.41)	**0.011** *****
**Correct visual acuity ^c^**			
Yes	1		
No	5.56	(1.04–29.77)	**0.045** *****
**Signs of depression ^d^**			
No	1		
Yes	19.33	(1.41–264.33)	**0.026** *****

^a^ Divorced individuals were excluded from the analysis due to the low number of cases. ^b^ Based on the numerical rating scale (NRS) (0–3, absence of pain; 4–10, presence of pain).^c^ Score ≥ 7 on the Jaeger chart (≥7). ^d^ Score ≥ 2 on the short version of the Yesavage scale. * *p* < 0.05.

## Data Availability

Not applicable.

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
