# Peer review of "Fear of Falling in Older Adults Treated at a Geriatric Day Hospital: Results from a Cross-Sectional Study"

_ijerph, 2022, doi:10.3390/ijerph19148504_

Round 1

Reviewer 1 Report

The manuscript structure is appropriate, but some important details need to be reviewed ( e.g. missing data), before being considered for publication.

Comments:

Introduction

1. Pg 1, lines 32-34. The loss of functional autonomy will likely have an impact on basic daily life activities. So, they are not really separate consequences. What about depression, can it be a consequence?

2. Pg. 1, line 42. Why being a female is a risk factor for FOF, according to literature? Is it linked to behaviour or because the number of females in older adult cohorts is higher than males?

Materials and Methods

3. Pg 2, line 72 “Illiterate individuals were excluded...” but in Table 1 “Level of education” they were included? So, were they excluded or not?

4. Pg2, lines 85-89.  Three suggestions/comments:

a. Consider replacing “sex” with “gender”

b. No need to describe each social-demographic variable [e.g. “(widowed, married, other); ”(no education, primary, secondary)”]  since they are already discriminated in Table 1.

c. separate scales/instruments from the above variables, different sentences since they are different variables.

5. Pg2, line 92.  On the one hand / on the other [hand]. The expression “On the other hand…” is used to introduce a statement that contrasts with a previous statement or presents a different point of view. So, “on the one hand” is missing.

Results

6. Pg 3. Table 1 and Table 2. The socio-demographic and clinical characteristics of the male participants are missing. It must be added.

7. Pg 4 line126. What does 37.97 means? “The median (…) ABC scale was 76.57 (37.97).”

8. Pg 4 line 127. “be afraid” instead of “have fear”

9. Pg4 lines 126 and 127. Both sentences “Of the 62 (…) 38.7% (95% CI 26.2%-51.2%)” are about the same result. Chose one.

10. Pg 4. Lines 131-138. Since the data from males (n=21) were missing from Table 2, I was not able to review the data reported.

11. Pg 4 Lines 131-138. Authors should be more attentive to reporting, for instance: “Regarding individuals with fear of falling, (…) did not present with pain. (…) Regarding individuals without fear of falling, (…) over 75% reported no pain”.

How many (%) were in the first case? Use the same metrics when describing the data (like it was done in the other variables), otherwise, it can be confusing to readers and misleading to others less attentive.

12. Pg 5. How was pain revealed to be a risk factor, since in both groups fear of falling vs without fear of falling, the majority of the participants reported no pain?

13. Pg 5 and 6 lines 209 “we were only able to analyze the influence of those determinants that were already described in the literature.” Explain what do you mean?

14. Pg 6. As the Authors stated FOF is defined as a decrease in usual daily activities caused by a self-reported fear of failing in people with and without a history of falls. However, the “history of falls” is a risk factor, also confirmed in this study. So, should it be included/stated in the definition of the syndrome? What is the author's opinion?

Author Response

We appreciate the opportunity to revise the manuscript entitled “Fear of Falling in Older Adults Treated at a Geriatric Day Hospital: Results From a Cross-Sectional Study” (ijerph-1753897). The manuscript has been modified to reflect the very helpful comments provided by yourself and the Reviewers. We provide responses point-by-point in this box and incorporated in the revised manuscript marked with “Track Changes” function in MS Word.

Introduction

  • Pg 1, lines 32-34. The loss of functional autonomy will likely have an impact on basic daily life activities. So, they are not really separate consequences. What about depression, can it be a consequence?

We thank the reviewer for the suggestion and have included modifications to the revised version of the manuscript.

  • Pg. 1, line 42. Why being a female is a risk factor for FOF, according to literature? Is it linked to behaviour or because the number of females in older adult cohorts is higher than males?

We thank the reviewer for this comment. Is not so usual to discuss the reasons concerned to this difference in the literature, but in the last years some authors explained it as a differentiated gender behavior: their theory is that females are more concerned about their health and is easiest for them to admit their fears related to fall, although men could have the same feeling. These possible causes are referenced in the discussion (lines 192-196). Related to the fact that the number of females is higher than males, we must had in mind that risk is computed as a ratio between incidence I each group (or odds, depending on the study design), and total female number and total male number are also included in each denominator.

Materials and Methods

  • Pg 2, line 72 “Illiterate individuals were excluded...” but in Table 1 “Level of education” they were included? So, were they excluded or not?

We thank the reviewer for the suggestion. Illiterates were excluded, understood as people who cannot read or write. In the descriptive table, the level of education differentiates those who can read and write but did not complete primary level. We have included modifications in table 1 and table 2 differentiating primary (not completed) and primary completed.

  • Pg2, lines 85-89. Three suggestions/comments:
  1. Consider replacing “sex” with “gender”
  2. No need to describe each social-demographic variable [e.g. “(widowed, married, other); ”(no education, primary, secondary)”] since they are already discriminated in Table 1.
  3. separate scales/instruments from the above variables, different sentences since they are different variables.

We thank the reviewer for the suggestion and have included modifications to the revised version of the manuscript.

  • Pg2, line 92. On the one hand / on the other [hand]. The expression “On the other hand…” is used to introduce a statement that contrasts with a previous statement or presents a different point of view. So, “on the one hand” is missing

We thank the reviewer for the suggestion and have included modifications to the revised version of the manuscript.

Results

  • Pg 3. Table 1 and Table 2. The socio-demographic and clinical characteristics of the male participants are missing. It must be added.

We thank the reviewer for the suggestion and have included modifications to the revised version of the manuscript.

  • Pg 4 line126. What does 37.97 means? “The median (…) ABC scale was 76.57 (37.97).”

We thank the reviewer for the suggestion and perhaps it is not clear enough, but the initials (IQR) are included at the beginning of the sentence and described in the methods section.

  • Pg 4 line 127. “be afraid” instead of “have fear”

Changes made to the to the revised version of the manuscript. Thanks for the suggestion.

  • Pg4 lines 126 and 127. Both sentences “Of the 62 (…) 38.7% (95% CI 26.2%-51.2%)” are about the same result. Choose one.

Thanks for the suggestion. Changes are made to the to the revised version of the manuscript.

  • Pg 4. Lines 131-138. Since the data from males (n=21) were missing from Table 2, I was not able to review the data reported.

We thank the reviewer for the suggestion. In the modified version, the data for men are already included in the gender variable.

  • Pg 4 Lines 131-138. Authors should be more attentive to reporting, for instance: “Regarding individuals with fear of falling, (…) did not present with pain. (…) Regarding individuals without fear of falling, (…) over 75% reported no pain”.

How many (%) were in the first case? Use the same metrics when describing the data (like it was done in the other variables), otherwise, it can be confusing to readers and misleading to others less attentive.

Changes made to the to the revised version of the manuscript. We thank the reviewer for the suggestion

  • Pg 5. How was pain revealed to be a risk factor, since in both groups fear of falling vs without fear of falling, the majority of the participants reported no pain?

 We thank the reviewer for the suggestion. Despite of the majority of participants reported no pain, in those who reported fear of falling the proportion that report pain are highest than in the no fear of falling group, and this difference, in its relative form (OR) are statistically significant.

  • Pg 5 and 6 lines 209 “we were only able to analyze the influence of those determinants that were already described in the literature.” Explain what do you mean?

We refer to the fact that a cross-sectional study cannot establish risk factors, as opposed to a longitudinal study where a causality can be established. We thank for the suggestion.

  • Pg 6. As the Authors stated FOF is defined as a decrease in usual daily activities caused by a self-reported fear of failing in people with and without a history of falls. However, the “history of falls” is a risk factor, also confirmed in this study. So, should it be included/stated in the definition of the syndrome? What is the author's opinion?

This point of view is very interesting.  Despite that the origin of the fear of falling syndrome was a post-fall syndrome with the evolution of the concept, it was observed that this behavior was also present in people without previous falls. If the previous falls experience had been included in the syndrome definition, we could not take in mind this interesting subset of individuals without history of falls.  Due that, we feel comfortable with the definition collected

Reviewer 2 Report

The authors conducted a cross-sectional study to estimate the prevalence and describe the main determinants of fear of falling in a sample of older adults attending a geriatric day hospital. The prevalence of fear of falling in this sample of 62 older individuals was 38.7%. In addition, previous falls, depression, pain, and poor visual acuity were associated with fear of falling after adjustment.

There are some comments.

Comments:

1.      Materials and Methods (Study Outcomes at Line 81-82 on page 2): The authors used the Activities-specific Balance Confidence (ABC) Scale to assess fear of falling. However, the rationale was unclear. Was the decision based on its better psychometric properties? In addition, I would suggest adding the comparisons between ABC and mFES in DISCUSSION.

2.      Materials and Methods (Study Outcomes at Line 83 on page 2): “An individual is considered to have fear of falling when his/her score is lower than 67 points”. However, this cutoff point was derived based on its ability to separate fallers from non-fallers rather than those with fear of falling from those without. Using this cutoff to define fear of falling is questionable.

3.      Materials and Methods (Study Design and Population at Line 70 on page 2): The authors recruited those who were able to walk 10 meters without stopping in this study. However, the reason is unclear. As stated in the INTRODUCTION (Line 57 on Page 2), “no specific data for day hospital patients are available—individuals who, although still living in the community and not requiring hospitalization, need specialized health care-.” How would the prevalence and determinant of fear of falling among those who attended a geriatric day hospital and could walk 10 meters without stopping fill the knowledge gap? If this study were part of a larger/prospective study, I would recommend a description here. Otherwise, I would suggest a discussion of the issue in DISCUSSION.

4.      Discussion: As stated in the INTRODUCTION (Line 43 on Page 1), chronic diseases are potential correlates of fear of falling. However, chronic diseases were not measured in this study. A discussion of this limitation in DISCUSSION is recommended.

5.      Discussion: The authors recruited a sample of older adults attending an ambulatory geriatric care center in Barcelona. However, it is unclear how representative this sample was. Were those attending that care center similar to those attending other care centers? A major limitation of this study is the issue of generalizability. I would recommend a discussion of this issue in DISCUSSION. 

Author Response

We appreciate the opportunity to revise the manuscript entitled “Fear of Falling in Older Adults Treated at a Geriatric Day Hospital: Results From a Cross-Sectional Study” (ijerph-1753897). The manuscript has been modified to reflect the very helpful comments provided by yourself and the Reviewers. We provide responses point-by-point in this box and incorporated in the revised manuscript marked with “Track Changes” function in MS Word.

  • Materials and Methods (Study Outcomes at Line 81-82 on page 2): The authors used the Activities-specific Balance Confidence (ABC) Scale to assess fear of falling. However, the rationale was unclear. Was the decision based on its better psychometric properties? In addition, I would suggest adding the comparisons between ABC and mFES in DISCUSSION.

We thank the reviewer for the suggestion. The Confidence and Balance in Specific Activities (ABC) scale is a more specific measure of a situation of balance confidence in not falling while performing these activities while other scales assess levels of concern for falling while performing activities of daily living. So, our interest was to pick up on that aspect of trust and not so much of worry. A rationale about de selection is comment in the discussion (lines 180-185).

ABC add six items related to the instrumental activities of daily living of the Falls Efficacy Scale (FES); modified version of the FES (mFES) expanding four items referring to outdoor activities. Alike, ABC are validated in Spanish but mFES is not validated.

Related to psychometric properties, ABC is one that has better reliability and validity compared to other scales that measure self-efficacy, 0.96 and 0.84 respectively. It has a sensitivity of 0.65, below the FES which has a score of 0.78. This has also been a reason for selecting the scale.

Modifications have also been made in the manuscript in materials and methods and in discussion.

  • Materials and Methods (Study Outcomes at Line 83 on page 2): “An individual is considered to have fear of falling when his/her score is lower than 67 points”. However, this cutoff point was derived based on its ability to separate fallers from non-fallers rather than those with fear of falling from those without. Using this cutoff to define fear of falling is questionable.

The ABC is a measure of the degree of confidence of balance in not falling while performing activities of daily living. Given that in our case the variable was not following a normal Distribution and, therefore, could not be treated as a continuous variable, it was transformed in a dichotomous variable with a cut-off mark of 67 points. It was considered that the individuals with a punctuation below this had a high degree of fear of falling because it is established that it marks the cut-off that discriminates fallers from non-fallers.

  • Materials and Methods (Study Design and Population at Line 70 on page 2): The authors recruited those who were able to walk 10 meters without stopping in this study. However, the reason is unclear. As stated in the INTRODUCTION (Line 57 on Page 2), “no specific data for day hospital patients are available—individuals who, although still living in the community and not requiring hospitalization, need specialized health care-.” How would the prevalence and determinant of fear of falling among those who attended a geriatric day hospital and could walk 10 meters without stopping fill the knowledge gap? If this study were part of a larger/prospective study, I would recommend a description here. Otherwise, I would suggest a discussion of the issue in DISCUSSION

The same inclusion and exclusion criteria have been used as in studies carried out with older people living in the community with the aim of seeing differences with this population group for which no information is available. We thank the reviewer for the suggestion and have included modifications to the revised version of the manuscript.

  • Discussion: As stated in the INTRODUCTION (Line 43 on Page 1), chronic diseases are potential correlates of fear of falling. However, chronic diseases were not measured in this study. A discussion of this limitation in DISCUSSION is recommended.

We thank the reviewer for the suggestion and have included modifications to the revised version of the manuscript.

  • Discussion: The authors recruited a sample of older adults attending an ambulatory geriatric care center in Barcelona. However, it is unclear how representative this sample was. Were those attending that care center similar to those attending other care centers? A major limitation of this study is the issue of generalizability. I would recommend a discussion of this issue in DISCUSSION.

In Catalonia, northeastern Spain, the Sociosanitary Master Plan will develop a range of resources between the hospital and the home to help the person regain their independence as quickly as possible, providing a safer transition and avoiding unnecessary admission to residences. Among the services offered includes Geriatric Day Hospitals, which is defined as an interdisciplinary day center, integrated into a hospital, where the frail elderly or geriatric patient, usually with a physical disability, goes to receive comprehensive treatment and geriatric assessment and later return home. This portfolio establishes both the profile of the person to be cared for and the services that must be offered to guarantee equitable care throughout the territory. The selected center is representative of geriatric day hospitals in urban area.

Likewise, have included modifications to the revised version of the manuscript and we thank for the improvement.

Round 2

Reviewer 2 Report

The authors have addressed the concerns in this version of the manuscript.